# OpenReview forum: "InfoCLIP++: A Multimodal Learning Framework with Multi-Granular Information-Theoretic Alignment and Adaptive Fusion"
_ICLR.cc/2026/Conference — ICLR 2026 Conference Withdrawn Submission_

### Official Review · Reviewer_JNRM · 2025-10-29

**Soundness:** 2
**Presentation:** 1
**Contribution:** 2
**Rating:** 2
**Confidence:** 3

**Summary:**

This paper proposes a new model for multi modal learning, which is called InfoCLIP++.
InfoCLIP++ integrates optimal transport, random feature approximation, hardware-aware optimization and other techniques.
The experiments demonstrate that InfoCLIP outperforms baselines on Zero-Shot Classification, Cross-modal retrieval, fine-grained alignment, and noise robustness.
Furthermore, InfoCLIP is 6.8 times faster than CLIP in calculations on FPGAs.

**Strengths:**

- The proposed framework achieves high practical performance. I like that it also outperforms baselines in terms of computational complexity.
- At least the combination of technologies in the framework proposed in this paper seems novel. Individual technologies may possess some degree of novelty.
- The experiment is evaluated from various perspectives.

**Weaknesses:**

- This paper is not well written, and its central argument remains unclear throughout.
Though Section 2 is the most crucial part of this paper, the connections and structure between subsections are unclear.
The connection of subsections seems like a list of bullet points, and the section seems to lack any in-depth exploration of the topics it aims to address.
For example, though this paper claims GPU acceleration and FPGA deployment, readers cannot utilize them or use them as a starting point for new ideas because they are scarcely explained.

- The current structure and content of the paper make it difficult to draw general or transferable insights for the broader machine learning community.
Since the work is primarily empirical, it would benefit from narrowing the focus and designing experiments that more directly support the central claim.
While some of the individual ideas appear interesting, the current presentation does not clearly convey their novelty or justify their effectiveness.
If the main contribution lies in achieving higher empirical performance rather than providing conceptual insights, the paper might be more suitable for an application-oriented conference.

- To make the paper easier to understand, I think it would be better to create a section explaining the ideas and overall picture before Section 3, and describe which section each idea is written in. Also, explaining how each technology was conceived, including comparisons with existing technologies, might make it easier to understand.
Theoretical results to support the claims will strengthen the paper. If difficult, toy problems or visualizations might help to support the claims.

**Questions:**

- What is the most significant contribution of this paper? How did you demonstrate it?
- Does this paper contain insights that will influence the broad field of machine learning?

---

### Official Review · Reviewer_Lt5K · 2025-11-01

**Soundness:** 3
**Presentation:** 2
**Contribution:** 3
**Rating:** 4
**Confidence:** 4

**Summary:**

This paper proposes InfoCLIP++, a multimodal foundation model that enhances CLIP by integrating multi-granular alignment, differentiable adaptive routing, and hardware-aware optimization. The framework performs semantic alignment at pixel, patch, and global levels via constrained optimal transport and Random-Feature HGR correlation, achieving fine-grained yet efficient cross-modal understanding. A differentiable routing mechanism enables adaptive token and modality pruning based on semantic importance, while quantized RF-HGR projections allow efficient deployment on GPUs and FPGAs. The method achieves significant improvements on standard benchmarks and large computational gains.

**Strengths:**

- Clearly identifies and addresses three long-standing challenges of CLIP-style models—coarse alignment, inefficiency, and noise sensitivity.
- The combination of constrained optimal transport and RF-HGR correlation offers a principled, information-theoretic approach to multi-scale alignment.
- Considerable performance gains across multiple benchmarks with both accuracy and efficiency improvements.

**Weaknesses:**

- The paper is very dense and sometimes difficult to follow; architectural details could be clearer.
- While the paper adopts an optimal transport  framework for pixel–text alignment, OT is theoretically defined between comparable probability distributions. Here, pixel features and textual tokens reside in heterogeneous embedding spaces, and it is unclear whether treating them as directly comparable distributions is conceptually justified. The semantic validity of such cross-modal transport mappings therefore remains uncertain.
- The modality-level routing mechanism assumes that modalities with globally lower activation scores contribute little and can be pruned. However, in multimodal fusion, redundancy across modalities often serves a complementary or stabilizing purpose—providing robustness under noise, occlusion, or modality failure. The current design does not clarify how it distinguishes between “redundant but useful” and “truly uninformative” modalities.
- Although the hardware-oriented acceleration enhance practicality, they do not appear to introduce substantial algorithmic or conceptual innovation beyond standard quantization and batching strategies.

**Questions:**

See above.

---

### Official Review · Reviewer_b1Av · 2025-11-02

**Soundness:** 2
**Presentation:** 1
**Contribution:** 2
**Rating:** 2
**Confidence:** 3

**Summary:**

This paper aims to enhance the CLIP model through three key aspects: (i) alignment granularity, (ii) computing efficiency, and (iii) robustness against noisy samples. The authors propose a hierarchical approach to alignment that involves pixel-word, token-attribution, and image-text correlation to address the first aspect. To tackle the second and third aspects, they introduce a differentiable adaptive routing mechanism that takes into account the quality of cross-modal alignment, thereby improving both computational efficiency and the model's robustness to noise.

**Strengths:**

The hierarchical designs address several crucial aspects of CLIP, enhancing both its alignment quality and computational efficiency.

**Weaknesses:**

- The presentation quality is poor:

    - It is difficult for readers to follow the content due to weak logical connections between the components, and there is no clear motivation provided for each specific design choice.

    - The equations are challenging to read. For instance, $T$ refers to the text in Fig. 1, while the transport matrix in Eq. 2. Additionally, there is no definition of $t$ in Eq. 11, which I assume refers to tokens. It is also unclear what the patches $p'$ aggregated with attributions are used for.

    - Fig. 1 is overly complex and confusing.

    - There are missing details: what are the implementation specifics for Tab. 3? How does the mIoU result of ADE20K achieve 70+?

- The effects of each component are unclear. Given the complexity of the proposed method, the ablation section should undergo significant revision to clarify the effects and learning dynamics of each design element.

- The claim that the proposed designs are scalable lacks support. I wonder whether the designs remain stable when applied to larger ViTs.

**Questions:**

please refer to the weakness part.

---

### Official Review · Reviewer_i3Fn · 2025-11-11

**Soundness:** 3
**Presentation:** 2
**Contribution:** 3
**Rating:** 6
**Confidence:** 3

**Summary:**

The paper targets three long-standing issues in CLIP-like multimodal learning: (i) coarse alignment (missed fine-grained correspondences), (ii) high compute for high-res inputs, and (iii) fragility to noise. InfoCLIP++ proposes an integrated framework with:
1) Multi-granular alignment: pixel-level constrained OT; patch/global alignment via RF-HGR, with an MI lower-bound motivation.
2) Differentiable Adaptive Routing (DAR) for token/modality pruning using agreement, saliency, and gradient criteria with Gumbel-Sigmoid.
3) Hardware-aware optimization for RF-HGR and a composite loss combining all pieces.

**Strengths:**

- Principled alignment across granularities: Pixel-wise OT + patch/global RF-HGR is a coherent design; the MI link via HGR (Eq. 8–9) gives a clean objective-level story for maximizing cross-modal information.


- End-to-end efficiency via DAR: Token/modality-level routing is differentiable and shows clear contribution to noise robustness (NRS +0.22 from base CLIP to full model) and efficiency (less FLOPs, reduced latency across GPU/CPU/FPGA).

- Empirical coverage and clarity: Results span zero-shot classification, cross-modal retrieval, segmentation transfer, and robustness.

- Comparative advantage over CLIP family: The reported ImageNet-1K/COCO improvements are sizable vs. CLIP/SigLIP/ALIGN and competitive with ALBEF/BLIP/BLIP-2, while adding efficiency knobs the latter lack.

**Weaknesses:**

- No Related Work Section.
Relation to broader “info-theoretic CLIP” literature:
There is growing work analyzing/optimizing CLIP via MI/correlation perspectives. A short related-work bridge to information-theoretic T2I/CLIP alignment papers would help situate RF-HGR’s role.


- Novelty vs. recent fine-grained alignment papers:
SmartCLIP (CVPR’25) formalizes identification guarantees and disentanglement for multi-granular alignment; InfoCLIP++’s pixel-OT + RF-HGR is strong but positioning against SmartCLIP-style theory is brief. What, precisely, is new: the RF-HGR + constrained OT + DAR combination, or improved approximations/efficiency? A direct comparison/discussion is warranted.

- RF-HGR approximation rigor and sensitivity:
The paper uses random Fourier features to approximate HGR; however, approximation error (ε) is only symbolically acknowledged in Eq. (9). Please report sensitivity to k, σ, ε and Sinkhorn iterations/ε and show accuracy/latency/variance trade-offs, especially given the hardware-aware pitch.


- Retrieval benchmarking details:
COCO R@1 = 74.5 is strong; please clarify backbone size, pretraining data, and retrieval protocol.

**Questions:**

- RF-HGR hyper-parameters: Provide curves for k (RF dimension), σ, and the hardware-aware quantization setting—how do accuracy and latency scale?
OpenReview

- DAR stability: How sensitive is routing to the temperature schedule / sparsity target ρ and the λA, λG, λE weights in Eq. (10)? Any failure modes (e.g., over-pruning text tokens on short captions)?
OpenReview

- SmartCLIP comparison: Can you add a head-to-head (classification/retrieval) and/or discuss theoretical differences (HGR-MI vs. identification guarantees)?


- Protocol parity: Confirm backbone sizes and pretraining corpora vs. ALBEF/BLIP/BLIP-2 in Table 1 comparisons.

---

### Note · Authors · 2025-11-12

I have read and agree with the venue's withdrawal policy on behalf of myself and my co-authors.